# Determinants of Outdoor Time in Children and Youth: A Systematic Review of Longitudinal and Intervention Studies

**DOI:** 10.3390/ijerph20021328

**Published:** 2023-01-11

**Authors:** Richard Larouche, Madeline Kleinfeld, Ulises Charles Rodriguez, Cheryl Hatten, Victoria Hecker, David R. Scott, Leanna Marie Brown, Ogochukwu K. Onyeso, Farzana Sadia, Hanako Shimamura

**Affiliations:** 1Faculty of Health Sciences, University of Lethbridge, Lethbridge, AB T1K 3M4, Canada; 2School of Human Services, Lethbridge College, Lethbridge, AB T1K 1L6, Canada; 3Library, University of Lethbridge, Lethbridge, AB T1K 3M4, Canada; 4Faculty of Applied Community Studies, Douglas College, Coquitlam, BC V3B 7X3, Canada

**Keywords:** outdoor play, social-ecological model, adolescents, physical activity, nature

## Abstract

Spending more time outdoors can improve children’s social and cognitive development, physical activity, and vision. Our systematic review summarized the determinants of outdoor time (OT) based on the social-ecological model. We searched nine databases: MEDLINE, APA PsycINFO, Web of Science, Cochrane Central Register of Controlled Trials (CENTRAL), CINAHL, SPORTDiscus, ERIC, SocINDEX, and ProQuest Dissertations and Theses. To be included, studies needed to be quantitative and longitudinal, include ≥1 potential determinant of OT among 0- to 17-year-olds, and be published in English, French, Japanese, or Spanish. We extracted the authors, publication year, country, design, sample size, OT measures, follow-up period, potential determinants, main results, and potential moderators or mediators. Fifty-five studies examining 119 potential determinants met the inclusion criteria. OT was consistently higher in warmer seasons and among participants reporting more OT at baseline. All three interventions that included both parent sessions and additional resources to promote OT (e.g., specific advice and community guides) were effective. COVID-19 restrictions and sun safety interventions discouraging midday outdoor activities led to less OT. The quality of evidence was rated as weak for 46 studies. Most potential determinants were examined in ≤3 studies; thus, more longitudinal studies are needed to enable stronger conclusions about the consistency of evidence and meta-analyses.

## 1. Introduction

Evidence from large systematic reviews indicates that children and youth who are more physically active reap multiple benefits, including better motor and cognitive development, greater cardiovascular fitness, reduced cardiometabolic risk, and improved bone health [1,2]. However, the majority of children and youth worldwide do not meet current physical activity (PA) guidelines [3,4]. For example, Guthold et al. [4] estimated that, among 1.6 million 11- to 17-year-olds from 146 countries, 77.6% of boys and 84.7% of girls were insufficiently active. This underscores a need for interventions targeting important determinants of PA. Systematic reviews have consistently identified positive associations between time spent outdoors and children’s PA [5,6,7,8]. Yet, previous research suggests that outdoor play (OP) has declined substantially over the last few decades, likely at the expense of increasing screen time [9,10], emphasizing a need to promote outdoor activities.

Given that COVID-19 restrictions were associated with a decline in PA and limited access to opportunities such as physical education and sports, promoting outdoor activities may be even more salient in pandemic and post-pandemic contexts [11]. Unfortunately, cross-sectional studies included in the scoping review by Paterson et al. [11] reported that children spent limited time outdoors during the pandemic.

Beyond PA, a growing body of evidence shows that spending more time outdoors is beneficial for social-emotional and cognitive development, psychosocial health, and vision [12,13,14]. Exposure to nature has been shown to prevent myopia [12], improve the management of attention-deficit/hyperactivity disorder [15,16], and foster pro-environmental behaviours among children [17]. Based on some of the abovementioned benefits, the Position Statement on Active Outdoor Play recommends that “*Access to active play in nature and outdoors—with its risks— is essential for healthy child development. We recommend increasing children’s opportunities for self-directed play outdoors in all settings—at home, at school, in child care, the community and nature*” [14]. It is worth noting that the phrases outdoor time (OT) and OP have been used interchangeably in the literature, even though the former concept is broader [18]. Specifically, the Play, Learn, and Teach Outdoors Network defined play as “Voluntary engagement in activity that is fun and/or rewarding and usually driven by intrinsic motivation” and OP as a form of play that occurs outdoors [18]. Conversely, OT refers to the amount of time spent outdoors [18], regardless of what people are doing while outdoors.

Although previous systematic reviews have examined the *correlates* of OT and/or OP [19,20,21], most included studies were cross-sectional, precluding conclusions about the direction of observed associations. One of these reviews focused only on parental correlates [19], and another focused only on the built environment [20]. In addition, two of these reviews excluded adolescents [19,21]. A systematic review that focuses specifically on longitudinal studies that can establish a temporal sequence would help identify key *determinants* that could be targeted in future interventions. Such information would be particularly useful for researchers, practitioners, and policymakers to develop evidence-informed interventions to increase OT.

To address these research gaps, we conducted a systematic review of longitudinal studies on the determinants of OT in children and youth (aged 0–17 years). Guided by the social-ecological model [22], we considered potential correlates representing the individual, interpersonal, community, built and natural environment, and policy levels of influence. In an effort to inform future interventions, we also examined the potential moderators and mediators of the determinants of OT.

## 2. Materials and Methods

We conducted this systematic review in accordance with the PRISMA 2020 statement [23] (see Appendix A for the PRISMA checklist). A health sciences librarian (DS) searched nine databases (MEDLINE, APA PsycINFO, Web of Science, the Cochrane Central Register of Controlled Trials (CENTRAL), CINAHL, SPORTDiscus, ERIC, SocINDEX, and ProQuest Dissertations and Theses) on 9 and 10 March 2021. The search was updated on 6 July 2022. Search terms were identified through consultations between the lead author (RL) and the librarian, a scan of the titles and subject headings of preliminary search results, and a review of the titles and abstracts of nine seed articles collected by RL. Elements of search strings developed for previously published reviews also informed the search strategy [24,25,26]. To optimize the precision of searches while maintaining high sensitivity, the search string was revised several times in accordance with feedback from RL. The search strategy was first developed for MEDLINE (Figure 1) and adapted for the eight other databases (Appendix A). When possible, subject headings from controlled vocabularies (e.g., MeSH) were used in the search. To increase sensitivity, concepts were also entered in the search string as keywords, with truncation (e.g., child*) and proximity operators (e.g., adj5) used when appropriate. Boolean operators connected subject headings and keywords, as shown in Figure 1 and Appendix A. No limits were placed on publication date, but language filters were applied to capture articles published in either English, French, Japanese, or Spanish. Searches of reference lists of previous systematic reviews [19,20,21] and included studies were completed by VH, MK, and FS. The review is registered in PROSPERO (ID: CRD42021243959).

See Appendix A for search strategies implemented in other databases.

### 2.1. Inclusion and Exclusion Criteria

To be eligible for the review, published studies needed to include (1) participants aged 0–17 years (or parents/guardians reporting on behalf of children); (2) a measure of OT; (3) data on at least one potential determinant in relation to OT; and (4) a longitudinal and quantitative design (i.e., including intervention studies, prospective studies, and retrospective studies). Studies published in English, French, Japanese, or Spanish were eligible. Eligible measures of OT included child- or parent-reported measures, direct observation, devices (e.g., accelerometers equipped with lux sensors), and any other relevant methods. No restrictions were placed on study location(s) or type of determinants examined. Studies that did not include data for children were excluded. Literature reviews, commentaries, editorials, cross-sectional studies, qualitative studies, and articles not in the aforementioned languages were excluded.

### 2.2. Screening Process

Titles and abstracts of each record identified were screened independently by two reviewers (MK, UCR, CH, VH, HS, LB, or OKO) trained by RL. Next, full-text copies of each record that passed the first stage of screening were obtained and screened independently by two research assistants. In this second stage, reasons for exclusion were collated in an Excel spreadsheet. Disagreements were resolved in discussions with RL.

### 2.3. Data Extraction

RL developed a data extraction form in Microsoft Excel. Data extracted included lead author, publication year, country, study design, analytical sample size, measures of OT, follow-up period, potential determinants examined, main results, and information about any moderators or mediators examined. For potential determinants, main results, and moderators or mediators, data about all reported longitudinal associations relevant to our review were extracted. For the first eight articles, data extraction was conducted independently by RL and all five reviewers, who then met via Zoom to discuss the extracted data. Subsequently, data extraction was conducted independently by two review team members. Any substantive disagreement was resolved in discussions with RL. When information to be extracted was missing, RL contacted the corresponding author of the original study via email. Any information that remained missing or unclear from the articles is identified as such in the results and tables.

### 2.4. Study Quality

We assessed study quality with the Effective Public Health Practice Project Tool (EPHPP) [27], a well-established method used in public health reviews that include few randomized controlled trials [28,29,30,31]. The tool addresses eight components: (1) selection bias; (2) study design; (3) confounders; (4) blinding; (5) validity and reliability of measurement tools; (6) withdrawals and drop-outs; (7) intervention integrity; and (8) analyses. Following recommended procedures, we graded the first six components as “strong”, “moderate”, or “weak” [27]. Next, we rated studies as “strong” if there were no “weak” ratings, “moderate” if there was only one “weak” rating, or “weak” if there were two or more “weak” ratings [27]. As in previous PA systematic reviews using the EPHPP, we made some adaptations to the tool [28,29,30,31]. First, for group-level interventions such as cluster randomized controlled trials, wherein all members of a school or preschool were assigned to a control or experimental group, we assessed participation rate at the group level for the first component [28,30]. Second, for the component on confounders, we expected studies to control for sex/gender, age, and socio-economic status, given that these variables were identified as correlates of PA and OT in previous reviews [5,7,19,21]. Third, like Dietz et al. [29], we considered the blinding component non-applicable in observational studies, as there are no interventions that participants/staff can be blinded to. After performing quality assessments independently for all articles, RL and MK met to discuss their assessments and resolve any discrepancies. When insufficient information was provided in the article to assess a component, we consulted previous articles or protocols from the study (when available) and/or contacted the author for correspondence. If no response was obtained, the component was conservatively rated as “weak”.

### 2.5. Summary of Findings

Due to the methodological heterogeneity of included studies (e.g., large differences in study designs, measures of OT, potential determinants, follow-up period, and analyses; see Table 1), we considered meta-analyses inappropriate. Instead, we present a narrative synthesis, and we tabulated measures of effect size (e.g., regression coefficients and odds ratios) and statistical significance as presented in the articles. To synthesize the consistency of associations investigated in ≥3 studies as an indicator of confidence in the body of evidence for a particular determinant, we adopted a classification system used in previous systematic reviews of correlates of PA and OT [8,21,32,33]. Variables positively or negatively associated with OT in ≥60% of studies were considered “consistent” correlates and coded as + or −. When 34–59% of associations were positive or negative, we coded the variable as (+) or (−), representing a “possible” correlate. Finally, when <34% of studies supported an association, we coded the variable as 0, indicating no relationship [8]. All studies that met inclusion criteria were included in the summary of findings. As per our PROSPERO record, we intended to stratify results by gender and age group; however, because most potential determinants were examined in a few studies, we were only able to investigate gender and age by age group.

## 3. Results

Figure 2 depicts the flow of the review process. Briefly, 4589 records were identified after the exclusion of duplicates. Of these, 4189 were excluded based on title and abstract screening. There were 3 full texts that could not be retrieved, and 346 were excluded based on inclusion/exclusion criteria, leaving 51 included articles. Three additional articles were included from twenty-six potentially relevant records identified by scanning the reference lists of included articles, and one was added from a narrative review of park prescription schemes [88]. Based on EPHPP guidelines, 16 of the 55 included papers were considered randomized controlled trials (RCTs) or controlled trials [12,37,38,39,44,45,47,53,57,59,60,64,65,67,77,78], 3 were quasi-experimental studies [36,54,58], 32 were prospective observational studies [34,35,40,41,42,43,46,48,49,50,51,52,55,56,62,63,66,68,69,70,71,72,73,74,75,80,81,82,83,84,85,86], 3 were retrospective longitudinal studies [61,76,87], and 1 was an uncontrolled pilot study [79]. Some studies combined data from control and experimental groups [35,40,82] and were considered observational because exposure to the intervention was not of substantive interest in the analyses. Table 1 summarizes the descriptive characteristics of the included studies categorized by age group at baseline (preschoolers (<5 years), children (5–11 years), and adolescents (12–17 years)) based on the mean, grade level, or midpoint of the reported age range. We classified studies that included distinct cohorts based on the highest mean age/grade reported. At baseline, the majority of cohorts had a mean age of <12 years. Only seven studies included a cohort of participants aged ≥ 12 years at baseline [81,82,83,84,85,86,87]. One study was conducted in boys only [74], and one was performed in girls only [82], whereas other studies included boys and girls. The sample size varied from 7 to 26,611, and the follow-up duration ranged from 8 weeks to 15 years. Almost all studies (*n* = 50) were conducted in high-income countries (e.g., USA (*n* = 21), Australia (*n* = 11), the Netherlands (*n* = 5), Denmark (*n* = 2), Singapore (*n* = 2), Canada (*n* = 1), Chile (*n* = 1), Finland (*n* = 1), Germany (*n* = 1), Israel (*n* = 1), Italy (*n* = 1), Spain (*n* = 1), and the UK (*n* = 1)), whereas 5 studies were conducted in upper-middle-income countries (China (*n* = 4) and Brazil (*n* = 1)). One study was conducted in eight European countries [80].

### 3.1. Determinants of OT

Table 2 summarizes the potential determinants of OT and the main findings for each individual study stratified by age group. Table 3 presents the summary of associations between potential determinants (organized by levels of influence of the social-ecological model) and OT. Overall, the included studies investigated 119 potential determinants representing the personal, interpersonal, community, built and natural environment, and policy levels. Table 3 is not stratified by age group, as most included studies focused on 5- to 11-year-olds, and few determinants were investigated in ≥3 studies, limiting our ability to draw conclusions about the consistency of associations. It is also noteworthy that only four studies assessed policies in relation to OT [57,63,74,76].

With twenty studies, age was the most frequently examined variable. Overall, the association between age and OT is equivocal, with six studies reporting an increase with age [34,39,40,45,49,71], five showing a decrease [42,66,80,83,85], five showing mixed/inconsistent findings [43,55,70,81,82], and four reporting no associations [41,46,62,73]. However, five of the six studies reporting an increase involved younger cohorts (<5 years at baseline) [34,39,40,45,49], one of the studies coded as “mixed” reported an increase from 12 to 18 months of age followed by a steady decline until the last follow-up at 5 years [43], and one reported no changes in their younger cohort (ages 5–6 years at baseline), but a decrease in their older cohort (ages 10–12) [55]. Conversely, the average age was ≥5 years for four of the five studies reporting a decline in OT [66,80,83,85]. Five of the eleven studies that examined the association between gender and OT found that boys spent more time outdoors [50,71,81,83,85], one found similar findings in older but not younger children [55], and five found no gender differences [34,41,46,70,73]. Notably, all three studies examining gender differences in cohorts beginning in adolescence found significant differences [81,83,85]. Only two studies examined sex, reporting no differences between males and females [39,43].

Of the six studies examining parental education as a potential determinant, two found that children with more educated parents accumulated more OT [39,52], one found the opposite [70], two found no associations [41,61], and one found that children of more educated fathers increased their OT over time, whereas the mother’s education was not associated with OT [43]. For household income, two studies found no association [43,61], and one study found that higher income was associated with lower odds of eliminating OP or exercise during the COVID-19 lockdown in Singapore [76]. All seven studies that examined seasonal differences found that children spent significantly more time outdoors in warmer seasons [45,52,54,55,69,71,75]. Similarly, all three studies that assessed whether OT at a previous time point predicted current OT (i.e., past behaviour) reported significant positive associations [41,70,78]. Exposure to an intervention that included both sessions with parents and additional resources promoting OP (e.g., specific advice and community guide) was consistently associated with more OT (*n* = 3 interventions) [36,37,58]. In contrast, exposure to a school-based sun safety intervention that discouraged OT around midday was associated with lower midday OT among intervention groups in two out of three studies [53,64,65]. Lastly, all three studies that investigated the effect of the implementation of COVID-19 restrictions found decreases in OT [63,74,76], and one of these studies also reported that OT returned to pre-COVID levels after restrictions were lifted [74].

The criteria for consistency were not satisfied for any other potential determinant. It is worth noting that two of the three school-based curricular interventions that aimed to increase OT did not yield significant differences between experimental and control groups. In general, interventions that primarily focused on other health behaviours (e.g., PA in general and obesity) were less effective, with only 2/6 showing positive results. However, small sample sizes may have limited researchers’ ability to detect significant intervention effects. For example, the study by Ford et al. [59] included 28 participants, and despite a medium-to-large increase in OT following the intervention (Cohen’s d = 0.71), the effect was not statistically significant (*p* = 0.057). Similarly, Christiana et al. [54] found that an outdoor PA prescription scheme at a single pediatric practice did not result in increased OT at the 3-month follow-up, but the intervention was very small (*n* = 32) and took place from August to December, and the measure of OT was very crude (frequency assessed with a Likert scale). Still, 70% of parents reported using intervention materials, and 44% believed that the prescription encouraged their child to participate in outdoor activities [54].

### 3.2. Moderators and Mediators

As summarized in Table 4, twelve studies examined potential moderators. Child age (*n* = 3), sex (*n* = 3), and gender (*n* = 2) were the only potential moderators examined in at least two studies. Age was a significant moderator in two studies [46,63]. First, Shah et al. [46] found an interaction between age and the *future* risk of myopia and OT. Before the age of four, there were no differences in OT between children who later became myopic and those who remained non-myopic. Then, from 4 to 8.5 years, there was a larger decline in OT among children who became myopic. Second, Li et al. [63] examined whether age, sex, and household income moderated the effect of adherence to four preventive public health measures for COVID-19 on OT. Limiting the number of visitors was associated with a significant decline in OT in children under five years of age, but not in older children [63]. Two of the three studies examining sex as a moderator found no evidence of moderation [38,63]. Conversely, van Grieken et al. [77] found that males in the intervention group had a non-significant decrease in OP, whereas females demonstrated a significant and meaningful increase (>30 min/day). Regarding gender, French et al. [83] found no evidence of moderation, whereas Miller [86] claimed that there was a significant moderation effect but did not report the direction and magnitude of the association. However, Miller [86] did report that youth with lower levels of parental monitoring perceived their neighbourhood as more supportive and reported more OT, whereas youth with high parental monitoring perceived their neighbourhood as more dangerous and reported less OT. These findings suggest that parental monitoring may be a response to a perceived lack of safety that could instill in youth concerns about neighbourhood safety.

Remmers et al. [71] examined parent-perceived responsibility towards their child’s PA as a potential moderator. They noted that parent-perceived neighbourhood functionality was associated with more OP in children of parents with high perceived responsibility, while among parents with low perceived responsibility, functionality was related to less OP. They also found that the association between traffic safety and OP was stronger when parents perceived high vs. low responsibility. They suggested that “*parents who feel responsible for the amount of their child PA may deliberately provide their child with the autonomy to play outside at spaces that they think are appropriate and safe*” [71]. Handy et al. [61] found that the presence of children aged 6–12 in the household moderated the relationship between living in a cul-de-sac and the frequency of OP. Specifically, the presence of cul-de-sacs was supportive of OT for younger children compared to older children. However, the relationship between eleven other characteristics and OT was not modified by the presence of younger children [61]. Examining income as a potential moderator, Li et al. [63] noted that practising physical distancing led to a significant decrease in OT among children from families earning ≥ CAD 80,000, but not in those earning less. Finally, Schneor et al. [74] observed a larger decrease in OT during a full vs. partial COVID-19 lockdown. Only one study planned to examine potential mediators; however, their intervention was not effective, so the criteria for demonstrating mediation were not satisfied [78].

### 3.3. Study Quality

Table 5 summarizes the results of our quality appraisal with the EPHPP tool. Overall, 9 studies were rated as “moderate” [34,41,42,51,68,71,77,78,87], and the remainder (*n* = 46) were rated as “weak”. The components most frequently rated as “weak” were data collection tools (*n* = 43), selection bias (*n* = 30), confounders (*n* = 19), blinding (*n* = 18), and withdrawals (*n* = 15). Because we excluded cross-sectional studies, all included studies were rated as “moderate” or “strong” for their study design. The components most commonly rated as “strong” were withdrawals (*n* = 22), confounders (*n* = 18), study design (*n* = 16), data collection tools (*n* = 6), selection bias (*n* = 4), and blinding (*n* = 2).
ijerph-20-01328-t002_Table 2Table 2Determinants of outdoor time in children and youth.Author (Year)Potential Determinants ExaminedMain Results**Studies Beginning in Early Childhood (<5 Years)**Arcury (2017) [34]Gender, age, people per bedroom, number of inappropriate media (having a TV in view at meals and having a TV in the child’s bedroom), number of age-appropriate toys, limiting screen time, frequency of visits to play spacesCompared to baseline, the mean time mothers estimated their child playing in the yard or park was 29.9 min/day greater at year 1 and 20.1 min/day greater at year 2 (both *p* = 0.001). For each additional month of age at baseline (B = 1.0 min/day, *p* = 0.049) and each age-appropriate toy (B = 12.3, *p* = 0.001), children spent more time playing in the yard or park. Each unit increase in the limiting screen time score was associated with less OP: B = −6.4, *p* = 0.016. Cameron (2019) [35]Influence of peer groups (i.e., partner, friends, mothers’ group, and family) on child’s nutrition, TV time, and PANo association between influence of any peer group and time spent outside (all *p* > 0.05).Davison (2011) [36]Exposure to community guide and group sessionsOdds of playing outdoors > 60 min per day for the intervention site at follow-up compared to baseline was OR = 1.68 (95% CI = 1.19–2.37, *p* = 0.003). The adjusted OR for the intervention site at follow-up vs. comparison sites was 2.79 (95% CI = 1.94–4.02, *p* < 0.001).Essery (2008) [37]Effect of newsletter or booklet intervention on child feeding practices and physical activityThere was a significant increase in OP reported by the newsletter (*p* < 0.01) and booklet (*p* < 0.01) groups between baseline and post-test.Händel (2017) [38]Healthy Start Intervention, focused on changing diet, PA, sleep, and stress managementParticipants in the intervention group spent more time on sports and outdoor activities combined at follow-up (intervention: 400 min/wk (95% CI: 341, 459) vs. control group: 321 min/wk (95% CI: 277, 366); *p* = 0.02). OP did not differ between groups post-intervention (intervention: 316 min/wk (95% CI: 264, 368) vs. 265 (95% CI: 209, 321); *p* = 0.19).Hesketh (2015) [39]Age, child sex, and mother’s educationOT increased from 25.7 to 90.0 min/day from 4 to 20 months of age (*p* < 0.001). Children of university-educated mothers engaged in more OT at all 3 time points (all *p* < 0.05).Hnatiuk (2013) [40]AgeOutdoor time increased from 46.93 ± 46.64 to 61.10 ± 48.35 min/week (*p* < 0.001).Honda-Barros (2019) [41]Age, gender, maternal education, school type (private vs. public), school shift (afternoon vs. morning), parent participation in PA with children, excess weight, and OP at baselineChildren who participated in PA with their parents were more likely to spend ≥60 min/day in OP (OR = 1.79; 95% CI = 1.27–2.54, *p* < 0.01). Children who spent ≥ 60 min/day in OP at baseline were more likely to maintain this behaviour after reaching school age (OR = 1.45; 95% CI = 1.02–2.07; *p* = 0.04). Children with excess weight at baseline engaged in less OP (OR = 0.56; 95% CI = 0.39–0.80; *p* < 0.01). Huang (2021) [42]AgeOver 2 years, the proportion of children who played outdoors ≥ 7 times a week decreased from 67.4 to 62.1%, and the proportion who played outdoors ≥ 60 min decreased from 53.3 to 38.8% (both *p* < 0.001).Li (2022) [43]Age, child sex, household size and income, pregnancy depression score, screen use, phone use, maternal and parental age, race, education level, and occupationOT varied significantly with age (*p* < 0.001), with an increase from 12 to 18 months followed by a gradual decrease. Children from older and more educated fathers had an increase in OT over time (*p* < 0.05). Lumeng (2017) [44]Exposure to 1 of 3 interventions: (1) Head Start program + Preschool Obesity Prevention Series (targeting obesity prevention behaviours) + Incredible Years Series (IYS) (program to improve children’s self-regulation); (2) Head Start + Preschool Obesity Prevention Series; or (3) Head Start onlyThere were no differences in OT between children assigned to different interventions (intervention 2 vs. 3: change from baseline = −0.08 h/day, *p* = 0.48; intervention 1 vs. 3: change from baseline = 0.12 h/day, *p* = 0.25; intervention 2 vs. 1: change from baseline = 0.19 h/day, *p* = 0.06).Sääkslahti (2004) [45]Age, season, and exposure to intervention (parents of children in the intervention group received information and concrete suggestions on how, when, and where to encourage their child’s PA)OP varied with intervention (*p* = 0.041), age (*p* = 0.016), and season (=12.72, *p* < 0.001). There were also combined relationships with age and season (*p* < 0.001), as well as intervention, age, and season (*p* < 0.001). The age-dependent increase was stronger in the intervention group. Children in the intervention group played more outdoors (*p* = 0.041) and less indoors (*p* = 0.049) than controls. Shah (2017) [46]Age, gender, future likely myopia, and number of myopic parentsGirls spent less time outdoors than boys (β = −0.04), but the difference was not significant (*p* = 0.14). Through the study period, children with one or two myopic parents spent an average of ~0.1 SD units per day less time outdoor than children whose parents were both non-myopic (*p* < 0.01). OT decreased with age, but the difference was not significant (β = 0.007; *p* = 0.073). Tandon (2019) [47]Exposure to interventions (Active Play! and Outdoor Play!)In the preschool childcare centres receiving the Active Play! intervention, increases in outdoor child-initiated activity (18.8 min/day; 95% CI: 12.6, 25.0; *p* < 0.001), teacher-led activity (2.5 min/day; 0.1, 4.9; *p* = 0.04), and total OT (21.4 min/day; 14.6, 28.3; *p* < 0.001) were found. In the Outdoor Play! intervention group, OT increased by 24 min/day (95% CI: 19.3, 28.6; *p* < 0.001). Outdoor child-initiated activity increased (23.8 min/day; 19.1, 28.4), and outdoor teacher-initiated activity did not change significantly. The only significant post-intervention difference between interventions favoured Active Play! and was for outdoor teacher-led time (2.6 min/day; 4.5, 0.7; *p* = 0.008).Thiering (2016) [48]Birth in a rural (Wesel) vs. urban area (Munich)Adolescents born in Wesel spent more time outside in the summer than those in Munich (χ^2^ = 46.94; *p* < 0.00001) ^a^ based on the frequencies reported in Table 1. Xu (2016) [49]Age, sleep patterns (bedtime, sleep duration, sleep latency, sleep time > 10 h/day, and waking at night)Over time, there was an increase in children’s mean OP time (*p* < 0.0001). **Studies beginning in childhood (5–11 years)**Avol (1998) [50]Gender and ambient ozone concentrationOn average, boys spent ~37 min longer outside in the spring than girls (*p* < 0.001) and ~22 min more outside in the summer (*p* = 0.04). Bacha (2010) [51]Parent-perceived neighbourhood safety (classified in tertiles)No difference in OT between tertiles of parent-perceived neighbourhood safety (*p* = 0.90).Bagordo (2017) [52]Season, parental education, and father’s occupational level70.3% of children played outdoors for >1 h/day at follow-up (spring) vs. 33% at baseline (winter) (*p* < 0.001). Children whose parents had <26 combined years of education were more likely (59.7% vs. 48.7%) to engage in >1 h/day of OP (*p* < 0.001). Children whose fathers had level III or IV occupations (service worker or unemployed, respectively) were more likely (57.4% vs. 48.1%) to engage in >1 h/day of OP (*p* < 0.001).Buller (2020) [53]Exposure to intervention on sun safetyIn schools where principals implemented sun safety practices, parents reported that children spent less time outdoors between 10 am and 4 pm over one week (mean = 14.78 vs. 16.32 h; *p* = 0.033).Christiana (2017) [54]Exposure to outdoor PA prescription intervention and seasonNo difference in frequency of OT between groups and frequency of achieving ≥60 min of outdoor PA (*p* ≥ 0.29). OT declined from baseline (August) to follow-up (November/December; *p* < 0.01); authors attributed this finding to seasonality.Cleland (2008) [55]Seasons, age, and genderOT was higher during warmer vs. cooler months at Time 1 and Time 2 (*p* < 0.01). OT was higher in older boys vs. older girls in both seasons at both time points. OT during warmer months declined between Time 1 and 2 among older boys (*p* < 0.01), OT on weekends in the warmer months declined among older girls (*p* < 0.01), and OT on weekdays in the colder months increased among older girls (*p* < 0.01).Cleland (2010) [56]Outdoor tendencies, indoor tendencies, parental encouragement, social opportunities, rules and restrictions, parental belief that child must be supervised when playing outside, parent report that there are no adults to supervise child while playing outside after school, dog ownership, number of siblings, yard size, home PA opportunities, access to local destinations, and weather as barrier (individual items: cold/dark in the winter; heat in the summer)For younger boys, “high” indoor tendencies were associated with less OT (−168 min/wk; 95% CI = −324, −13), while “high” social opportunities were associated with more OT over 5 years (170 min/wk; 95% CI = 26, 314). Among older boys, higher indoor tendencies (“medium”: −215 min/wk; 95% CI = −311, −119; “high”: −324; 95% CI = −472, −176) and a lack of adult supervision (−47 min/wk; 95% CI = −91, −3) were associated with less OT, while “high” outdoor tendencies were associated with more OT over 5 years (123 min/wk; 95% CI = 40, 207). Among younger girls, higher indoor tendencies were associated with less OT (“medium”: −188 min/wk; 95% CI = −356, −21; “high”: −247 min/wk; 95% CI = −374, −120), while “high” parental encouragement was associated with more OT over 5 years (234 min/wk; 95% CI = 30, 438). Among older girls, “medium” outdoor tendencies (200 min/wk; 95% CI = 27, 374) and “high” parental encouragement were associated with more OT (151 min/wk; 95% CI = 67, 236), while a lack of adult supervision was associated with less OT (−34 min/wk; 95% CI= −60, −9). Cortinez-O’Ryan (2017) [57]Exposure to an evening street closure intervention (twice a week for 12 weeks)There were significant increases in median number of weekdays with OP (from 2 to 3; *p* < 0.01), after-school OP time (from 60 to 90 min; *p* = 0.02), and weekly after-school OP time in the experimental neighbourhood (from 120 to 300; *p* = 0.01). No changes were observed in the control neighbourhood. Flynn (2017) [58]Exposure to intervention: family resource workbook and 3 group sessionsDuring the program, families increased their time spent being active together by an average of 111 min/week above baseline. Outdoor PA time was higher than baseline in 3 out of 4 weeks (*p* < 0.05). Mean length of a family outdoor PA bout was significantly greater than at baseline for all 4 weeks of the program. Families increased the mean length of their family outdoor PA bouts by ~41 min/bout. Frequency of family outdoor PA bouts did not change significantly.Ford (2002) [59]Behavioural vs. counselling (control) intervention to reduce children’s television viewingCompared to the control group, the behavioural intervention led to a medium-to-large increase in OT (change of 1.0 ± 5.9 vs. −4.7 ± 9.4 h/wk; Cohen’s d = 0.71); however, this difference was not statistically significant (*p* = 0.057).Gerards (2015) [60]Exposure to the “Lifestyle Triple P” interventionThe increase in OT between baseline and first follow-up (4 months) was not significant (B = 2.85; 95% CI= −0.16, 5.86; *p* = 0.063; Cohen’s d = 0.56). Significant increase was found at the final (12 month) follow-up (B = 1.94; 95% CI = 0.04, 3.84; *p* < 0.05; Cohen’s d = 0.55). Handy (2008) [61]Parental preference for and perceptions of neighbourhood characteristics ((1) nearby amenities; (2) neighbourhood upkeep; (3) large back yard; (4) large front yard; (5) living in a cul-de-sac rather than on a through street; (6) low traffic on neighbourhood streets; (7) parks and open spaces nearby; (8) sidewalks through neighbourhood; (9) lots of interaction among neighbours; (10) lots of people out and about in the neighbourhood; (11) low crime rate in neighbourhood; (12) safe neighbourhood for children to play). Perceived changes in abovementioned characteristics, age of children in the household, parental education, household income, changes in household size, changes in number of children in household, changes in income, type of housing (apartment vs. other) suburban vs. traditional neighbourhoodAfter residential relocation, 52.7% of parents reported no change in OP frequency, 15.7% reported a decline, and 31.5% reported an increase. Preference for a safe neighbourhood for kids to play was associated with more OP (β = 0.147; *p* = 0.028). Changes in 4 perceived neighbourhood characteristics were also associated with more OP: cul-de-sac interacted with the presence of children ages 6–12 years (β = 0.170 *p* = 0.014), large front yards (β = 0.200; *p* = 0.005), low crime (β = 0.290; *p* = 0.002), and interaction among neighbors (β = 0.189; *p* = 0.008). Parents with children aged 12–16 were more likely to report no change in OP vs. those with younger children. Households with children aged 5–12 were more likely to report an increase in OP (β = 0.234; *p* = 0.001) than households with older children. He (2015) [12]Exposure to outdoor play intervention (additional outdoor activity class; 40 min/school day)No difference in OT between children in intervention and control schools at baseline and 1-, 2-, and 3-year follow-ups (all *p* > 0.20).Kemp (2022) [62]AgeTime spent in “other outdoor/nature activities” (the time use category that included OT) did not change with age (all *p* > 0.05).Li (2021) [63]COVID-19 (number of days/week that children practised 4 preventive public health measures)For each additional day/week that children adhered to public health measures, OT decreased by 17.2 min/day in the unadjusted model (95% CI = −22.07, −12.40; *p* < 0.001) and by 12.5 min/day in the adjusted model (95% CI = −18.25, −6.79; *p* < 0.001). Adherence to each individual measure was associated with less OT in both the unadjusted and adjusted models (*p* < 0.05), except for limiting the number of visitors.Milne (2000) [64]Exposure to school-based multicomponent intervention with specially designed curriculum (“medium” intervention) vs. exposure to multicomponent intervention plus program materials over the summer holidays and low-cost sun-protective swimwear (“high” intervention) compared to standard health curriculum (“control”)Adjusted mean OT during the summer holidays between 11 am and 2 pm was highest in the control group (28.4 h) vs. the high intervention group (22.3 h) and the moderate group (24.1 h) (*p* = 0.01). Children in the moderate group tended to spend less time outside in both periods. Adjusted mean OT between 8 am and 4 pm was 111 h for the control group (95% CI: 103.9, 118.5), 113 h for the moderate group (104.6, 121.6), and 108.7 h for the high group (99.4, 118.5), with no differences between groups (*p* = 0.8).Milne (2007) [65]School-based sun protection curriculum over 4 years; children were encouraged to reduce sun exposure by staying indoors during the middle of the day, when solar ultraviolet radiation is highest, and to protect themselves when outdoors by using shade, clothing, hats, and sunscreenThe median OT in each group (control, “moderate” intervention, and “high” intervention) was similar after 2 years. There was no association between study group and total OT at either age 10 or age 12.Nigg (2021) [66]Age, OP (past behaviour), MVPA, TV, and computer/gaming time in previous survey wavesOP decreased from 5.93 ± 1.43 days/week at Time 1 to 1.14 ± 1.85 days/week at Time 3. OP at Time 1 was associated with more OP at Time 2, which was positively associated with OP at Time 3 (*p* < 0.05). Ngo (2009) [67]Exposure to intervention: structured weekend outdoor activities and incentives for children to increase their daily steps via pedometersAt the 6-month follow-up, a mean of 14.75 h/wk of OT was reported in the questionnaire for the intervention group vs. 12.40 h/wk for the control group (*p* = 0.04). At the 9-month follow-up, parents in the intervention and control groups reported 15.95 vs. 14.34 h/wk outdoors (*p* = 0.29). Mean OT from the diary was 6.98 h/wk and 7.93 h/wk for the control and intervention groups, respectively (*p* = 0.12).Nordvall-Lassen (2018) [68]“Moderate” preterm (32–36 weeks of gestation) vs. term birthNo difference in odds of reporting different weekly durations of OT based on birth status (OR for 4–6 h = 1.13 (95% CI = 0.59–1.74), OR for 7–13 h = 1.14 (0.61–1.70), OR for 14–20 h = 1.15 (0.52–1.67), and OR for 21–60 h = 1.20 (0.35–2.22)).Ostrin (2018) [69]Seasons (spring, summer, and fall) and parental outdoor timeChildren were more exposed to outdoor light (lux) in the summer (110.5 ± 45.8 min/day) vs. spring (94.2 ± 30.4 min/day) or fall (72.2 ± 31.0 min/day, *p* < 0.0001). Children received the highest mean daily light exposure during the summer vs. spring and fall seasons (*p* < 0.0001). Parent and child OT were significantly correlated (r = 0.76, *p* = 0.0002).Remmers (2014a) [70]Gender, age, and parental and environmental factors, including accessibility of PA-related places, attitude towards child PA, concern regarding child PA, restriction of screen time, social capital, functionality, traffic safety, attractiveness, perceived responsibility, pressure, and monitoringChildren spent on average ~60 more minutes in OP per week at 7 vs. 5 years of age (both boys and girls) (*p* < 0.01). At both time points, boys spent significantly more time in OP than girls (*p* < 0.01), and there were significant differences in OP duration between all seasons (*p* < 0.01; but season was examined as a random effect, and the direction of association was not reported). Significant regression coefficients (β) for parental factors were accessibility of PA-related places within 10 min walking distance of home with 0.05 (95% CI = 0.01, 0.09), positive parental attitude towards child PA with 0.09 (0.05, 0.13), concern regarding child PA with −0.04 (−0.09, −0.001), restriction of screen time with −0.21 (−0.26, −0.17), and social capital with 0.07 (0.03, 0.11). Remmers (2014b) [71]Socio-demographic characteristics (child age, gender, ethnicity, and BMI and parental age, ethnicity, BMI, and education), family environment (parental attitude, family attitude, perceived difficulty, habit strength and intention to improve OP, presence of rules, presence of monitoring, presence of active encouragement, and child autonomy), and parent-perceived physical environment (safety perception during daytime and evenings, friendliness for children, attractiveness for children, and safety of OP without supervision)Parents with high vs. low education reported that their child played outside 28.40 min/day less (95% CI = −55.66, −1.14) at age 7. Parents who indicated difficulty towards improving OP reported 22.11 (−33.41, −10.81) less minutes of OP. Parents with a habit towards improving OP (23.99; 95% CI = 14.61, 33.61), the presence of rules regarding OP (16.46; 95% CI = 9.26 to 23.67), and modelling from the respondent’s partner (1.85; 95% CI = 0.27, 3.42) were associated with more min/day of OP. Parental active encouragement of OP at baseline was associated with 8.91 (−17.33 to −0.48) less minutes of OP at age 7. Higher child age was related to more OP at baseline, but this attenuated significantly over time (*p_[interaction]_* < 0.01). Sadeh-Sharvit (2020) [72]Exposure to intervention (online 6-session parent-based prevention program after bariatric surgery)Children spent less time outdoors at follow-up as reported by parents with bariatric surgery (baseline: 125.63 (58.88) min/day; follow-up: 103.00 (61.40); Hedges’ g = 0.36 (−0.64, 1.35)) or partners (baseline: 102.86 (58.71) min/day; follow-up: 97.50 (102.10); Hedges’ g = 0.06 (−1.05, 1.18)). As 95% CIs cross 0, differences were not significant.Sanchez-Tocino (2019) [73]Age and genderThere were no significant differences in hours spent on outdoors activities by age or gender (*p* > 0.05).Schneor (2021) [74]COVID-19 restrictionsDaily OT decreased from 1.8 ± 1.0 h to 0.7 ± 0.7 h (*p* = 0.001). In the subsample followed up after restrictions were removed, OT returned to pre-pandemic levels (1.8 ± 0.8 h).Shepherd-Baniga (2014) [75]Farmworker vs. non-farmworker status and agricultural seasons (thinning vs. pre-thinning)Children spend 9 h/wk more outside during the thinning (summer) vs. pre-thinning season (spring) (95% CI: −13.0, −5.1, *p* < 0.001). In the thinning season, mean OT was 30.2 ± 20.8 h/wk for farmworker children vs. 24.2 ± 16.7 h/wk for non-farmworker children (*p* = 0.004).Sum (2022) [76]COVID-19 restrictions and household income64.1% of parents reported significant decreases in OP or exercise due to the COVID-19 lockdown (*p* < 0.001). Each 1,000-Singapore-dollar decrease in income before the lockdown was associated with higher odds of reporting the elimination of all OP or exercise (OR = 1.09; 95% CI = 1.01, 1.19; *p* = 0.03).Van Griecken (2014) [77]Exposure to a healthy lifestyle counselling intervention to parents of overweight 5-year-oldsProportion of children playing outside ≥ 1 h/day decreased in the intervention (93.3 to 77.1%; *p* < 0.001) and control groups (94.3 to 77.1%; *p* < 0.001), with no difference between groups (OR = 1.11; 95% CI = 0.60, 2.06). There was no significant difference in OP, expressed in min/day, at follow-up (β = 8.22; 95% CI = −15.77, 32.22) between groups.Van Stralen (2012) [78]Exposure to JUMP-in school-based intervention targeting sports participation and outdoor play. Also examined the effect of OP at baseline and child-perceived pros and cons, social pressure, social support, social modelling, self-efficacy, planning skills, barriers, enjoyment, and habit strength related to OPNo significant intervention effect in the weekly frequency of OP (B= −0.30; 95% CI= −0.79, 0.19). Significant positive associations were found between social support (b = 0.04; 95% CI: 0.01–0.08), self-efficacy (b = 0.15; 95% CI: 0.00–0.30), enjoyment (b = 0.21; 95% CI: 0.14–0.28), and habit strength (b = 0.38; 95% CI: 0.18–0.58) and OP. In their Figure 3, the authors show that OP at Time 1 (b = 0.17) and planning skills (b = 0.15) were significantly associated with more OP at Time 2 (but did not provide the *p*-value). Walker (2021) [79]Participation in a child-centred play therapy interventionThere were no differences in OT on weekdays and weekend days between baseline and the end of the intervention (all *p* > 0.20)Wolters (2022) [80]AgeOT was higher at baseline vs. the last follow-up: 2.41 ± 1.39 vs. 1.80 ± 1.29 h/day (*t* = 13.63; *p* < 0.0001 based on the values reported in Table 2) ^b^**Studies beginning in adolescence (12–17 years)**Dunton (2007) [81]Gender, age, time of week, and seasonCompared with girls, boys were more likely to report exercising in outdoor settings (*p* = 0.002) and walking in outdoor settings (*p* < 0.001). Walking in an outdoor setting decreased during high school (7% per year, *p* = 0.019), but outdoor exercising did not (p = 0.189). Students were more likely to exercise or walk outdoors on weekend days vs. weekdays (*p* < 0.001). Students were more likely to walk or exercise outdoors in the fall and spring seasons vs. the winter (all *p* < 0.05) and to walk outdoors in the spring vs. fall season (*p* = 0.010).Evenson (2018) [82]AgeThe number of park visits identified by GPS during the 6-day monitoring period increased from 73 to 83 (*p* < 0.02). Mean duration of park visits decreased from 63.9 to 38.4 min (*p* < 0.03). French (2013) [83]Age, ethnicity, and genderIn the young cohort, OT decreased by just over 1 h/wk from baseline to follow-up, accompanied by a decline in outdoor leisure (both *p* ≤ 0.001). Time spent on organized outdoor sports increased (*p* < 0.0001). In the older cohort, there was a significant decrease in OT and outdoor sporting activities (*p* < 0.0001), but not in outdoor leisure time (*p* = 0.06). Boys spent ~2.5 h/wk more outdoors than girls in both cohorts at baseline and follow-up (all *p* < 0.0001). The decrease in OT between baseline and follow-up was significant for girls in both cohorts (younger, *p* = 0.006; older *p* < 0.0001) and for boys in the older cohort (*p* = 0.001), but not in the younger cohort (*p* = 0.052). The decline in OT with age was seen in European Caucasian participants (younger cohort: *p* = 0.001; older cohort: *p* < 0.0001), but not in East Asian participants (younger cohort: *p* = 0.7; older cohort: *p* = 0.07).Gopinath (2013) [84]Birth weight (categorized in quartiles)In 12-year-olds, an increase in outdoor PA (~1 h/wk) was observed with increasing birth weight after adjustment for covariates (from the lowest to highest quartile; *p*_trend_ = 0.02). Among 17- to 18-year-olds, higher birth weight was associated with higher outdoor PA (~1 h/wk, *p* = 0.04). In multivariable models, each SD (573.5 g) increase in birth weight was associated with a 15 min/wk increase in outdoor PA (*p* = 0.01). Twelve-year-olds in the high- vs. very-low-birth-weight group (>4000 vs. <2000 g) spent ~1.3 h/wk more in outdoor PA (*p*_trend_ = 0.02).Lin (2017) [85]Age and genderAmong all students combined, there was a decrease in leisure OT (8.5 ± 7.7 vs. 9.9 ± 7.0 h/wk, *p* = 0.02). For primary students, there was an increase in outdoor sports time (3.5 ± 4.3 vs. 2.4 ± 3.0 h/wk, *p* = 0.02). For secondary students, there was a non-significant decrease in total OT (10.9 ± 8.8 vs. 12.7 ± 9.8 h/wk, *p* = 0.09). Girls spent less time outdoors than boys (baseline: 11.9 ± 7.9 vs. 14.4 ± 9.5 h/wk, *p* = 0.03; follow-up: 10.5 ± 8.2 vs. 13.9 ± 9.6 h/wk, *p* = 0.005) and less time in outdoor sports (baseline: 2.4 ± 2.8 vs. 4.0 ± 5.2 h/wk, *p* = 0.006; follow-up: 2.3 ± 3.1 vs. 5.1 ± 5.0 h/wk, *p* < 0.001).Miller (2017) [86]Parental perceptions of neighbourhood danger, perceived neighbourhood support, parental monitoring (in general), and OT at baseline There was a positive correlation between the percent time spent outside at Times 1 and 2 (r = 0.480). Increased levels of parental monitoring at Time 1 was associated with increased OT at Time 2 (coeff = 0.7508, *p* = 0.0109). Watowicz (2012) [87]Recent parental weight loss surgeryControl group participants were significantly more likely than those whose parents underwent weight loss surgery to report ≥1 h/day of OP (55.8 vs. 31.6%, *p* = 0.01). Note: When potential determinants are not mentioned in the third column, this means that they were not significantly associated with outdoor time. B = unstandardized regression coefficient; β = standardized regression coefficient; CI = confidence interval; coeff = coefficient; OP = outdoor play; OR = odds ratio; OT = outdoor time; PA = physical activity; SE = standard error. ^a^ Chi-square test performed by the review team based on reported frequencies using an online calculator (https://www.socscistatistics.com/tests/chisquare2/default2.aspx, accessed on 1 December 2022). ^b^
*t*-test performed by the review team based on reported means and standard deviations using an online calculator (https://www.graphpad.com/quickcalcs/ttest1/, accessed on 1 December 2022).
ijerph-20-01328-t003_Table 3Table 3Summary of the determinants of outdoor time in children and youth.Variable Level of InfluenceNumber of StudiesPositive (%)Negative (%)Mixed (%)Null (%)Summary CodeChild sex (male)Individual20 (0%)0 (0%)0 (0%)2 (100%)N/AChild gender (boy)Individual115 (45.5%)0 (0%)1 (9%)5 (45.5%)(+)Child age (older)Individual206 (30%)5 (25%)5 (25%)4 (20%)EquivocalChild ethnicity (dominant vs. other)Individual20 (0%)1 (50%)0 (0%)1 (50%)N/ABirth weight (higher)Individual11 (100%)0 (0%)0 (0%)0 (0%)N/APreterm birth (yes vs. no)Individual10 (0%)0 (0%)0 (0%)1 (100%)N/AChild weight status (overweight/higher BMI vs. normal weight)Individual20 (0%)1 (50%)0 (0%)1 (50%)N/AChild moderate- to vigorous-intensity PA (higher)Individual10 (0%)0 (0%)0 (0%)1 (100%)N/AChild screen time (higher)Individual20 (0%)0 (0%)0 (0%)2 (100%)N/ANumber of age-appropriate toys (higher)Individual11 (100%)0 (0%)0 (0%)0 (0%)N/AFrequency of visits to play spaces (higher)Individual10 (0%)0 (0%)0 (0%)1 (100%)N/AOutdoor tendencies score (higher)Individual10 (0%)0 (0%)1 (100%)0 (0%)N/AIndoor tendencies score (higher)Individual10 (0%)1 (100%)0 (0%)0 (0%)N/AOP at baseline (higher)Individual33 (100%)0 (0%)0 (0%)0 (0%)+Child-perceived neighbourhood danger (greater danger)Individual10 (0%)0 (0%)0 (0%)1 (100%)N/AChild autonomy (higher)Individual10 (0%)0 (0%)0 (0%)1 (100%)N/AChild-perceived pros and cons of OP (higher)Individual10 (0%)0 (0%)0 (0%)1 (100%)N/AChild-perceived social support for OP (higher)Individual11 (100%)0 (0%)0 (0%)0 (0%)N/AChild-perceived social modelling of OP (higher)Individual10 (0%)0 (0%)0 (0%)1 (100%)N/AChild-perceived self-efficacy for OP (higher)Individual11 (100%)0 (0%)0 (0%)0 (0%)N/AChild-perceived habit strength for OP (higher)Individual11 (100%)0 (0%)0 (0%)0 (0%)N/AChild planning skills for OP (higher)Individual11 (100%)0 (0%)0 (0%)0 (0%)N/AChild-perceived barriers to OP (higher)Individual10 (0%)0 (0%)0 (0%)1 (100%)N/AChild enjoyment of OP (higher)Individual11 (100%)0 (0%)0 (0%)0 (0%)N/AChild likely to develop myopia in the future (yes vs. no)Individual10 (0%)0 (0%)0 (0%)1 (100%)N/AChild sleep patternsIndividual10 (0%)0 (0%)0 (0%)1 (100%)N/AParticipation in a child-centred play therapy interventionIndividual10 (0%)0 (0%)0 (0%)1 (100%)N/AParent-perceived neighbourhood safety (safer)Interpersonal20 (0%)0 (0%)0 (0%)2 (100%)N/AParental education (higher vs. lower)Interpersonal62 (33.3%)1 (16.7%)1 (16.7%)2 (33.3%)EquivocalHousehold income (higher)Interpersonal31 (33.3%)0 (0%)0 (0%)2 (66.7%)0Father’s occupation (service workers/unemployed vs. higher class)Interpersonal11 (100%)0 (0%)0 (0%)0 (0%)N/AParental occupationInterpersonal10 (0%)0 (0%)0 (0%)1 (100%)N/AFamily sizeInterpersonal10 (0%)0 (0%)0 (0%)1 (100%)N/AChanges in household income (increase)Interpersonal10 (0%)0 (0%)0 (0%)1 (100%)N/AChanges in household size (increase)Interpersonal10 (0%)0 (0%)0 (0%)1 (100%)N/AChanges in the number of children in household (increase)Interpersonal11 (100%)0 (0%)0 (0%)0 (0%)N/AType of housing (apartment vs. other)Interpersonal10 (0%)0 (0%)0 (0%)1 (100%)N/ANumber of people per bedroom (higher)Interpersonal10 (0%)0 (0%)0 (0%)1 (100%)N/ANumber of “inappropriate” media (higher)Interpersonal10 (0%)0 (0%)0 (0%)1 (100%)N/ALimiting screen time (more restriction)Interpersonal20 (0%)2 (100%)0 (0%)0 (0%)N/ARules and restrictions related to OP (more vs. less)Interpersonal21 (50%)0 (0%)0 (0%)1 (50%)N/AParental concern with child’s PA (more concern)Interpersonal10 (0%)1 (100%)0 (0%)0 (0%)N/AParental attitude towards child’s PA (more positive)Interpersonal11 (100%)0 (0%)0 (0%)0 (0%)N/AParental and family attitude towards OP (more positive)Interpersonal10 (0%)0 (0%)0 (0%)1 (100%)N/APerceived difficulty of increasing OP (higher)Interpersonal10 (0%)1 (100%)0 (0%)0 (0%)N/AParental habit strength to improve OP (higher)Interpersonal11 (100%)0 (0%)0 (0%)0 (0%)N/AParental intention to improve OP (higher)Interpersonal10 (0%)0 (0%)1 (100%)0 (0%)N/AParental encouragement of OP (more encouragement)Interpersonal20 (0%)1 (50%)1 (50%)0 (0%)N/AParent participation in PA with children (higher)Interpersonal11 (100%)0 (0%)0 (0%)0 (0%)N/AParent OT (higher)Interpersonal11 (100%)0 (0%)0 (0%)0 (0%)N/AParent age (higher)Interpersonal20 (0%)0 (0%)1 (50%)1 (50%)N/AParent BMI (higher)Interpersonal10 (0%)0 (0%)0 (0%)1 (100%)N/AParent race/ethnicity (dominant vs. other)Interpersonal20 (0%)0 (0%)0 (0%)2 (100%)N/ASocial opportunities score (higher)Interpersonal10 (0%)0 (0%)1 (100%)0 (0%)N/ADog ownership (yes vs. no)Interpersonal10 (0%)0 (0%)0 (0%)1 (100%)N/ANumber of siblings (higher)Interpersonal20 (0%)0 (0%)0 (0%)2 (100%)N/AHome PA opportunities (higher)Interpersonal10 (0%)0 (0%)0 (0%)1 (100%)N/AParental belief that child must be supervised when playing outside (higher agreement)Interpersonal10 (0%)0 (0%)0 (0%)1 (100%)N/AParent report that there are no adults to supervise child while playing outside after school (higher agreement)Interpersonal10 (0%)0 (0%)1 (100%)0 (0%)N/APerceived influence of peer groups (i.e., partner, friends, mother, and family) on child’s nutrition, TV time, and PA (higher)Interpersonal10 (0%)0 (0%)0 (0%)1 (100%)N/AParental preference for access to nearby amenities (higher)Interpersonal10 (0%)0 (0%)0 (0%)1 (100%)N/AParental preference for high level of neighbourhood upkeep (higher)Interpersonal10 (0%)0 (0%)0 (0%)1 (100%)N/AParental preference for large back or front yard (higher)Interpersonal10 (0%)0 (0%)0 (0%)1 (100%)N/AParental preference for living in a cul-de-sac vs. on a through street (higher)Interpersonal10 (0%)0 (0%)0 (0%)1 (100%)N/AParental preference for low traffic on neighbourhood streets (higher)Interpersonal10 (0%)0 (0%)0 (0%)1 (100%)N/AParental preference for parks and open spaces nearby (higher)Interpersonal10 (0%)0 (0%)0 (0%)1 (100%)N/AParental preference for sidewalks throughout neighbourhood (higher)Interpersonal10 (0%)0 (0%)0 (0%)1 (100%)N/AParental preference for lots of interaction among neighbours or lots of people out and about in the neighbourhood (higher)Interpersonal10 (0%)0 (0%)0 (0%)1 (100%)N/AParental preference for low crime rate in neighbourhood (higher)Interpersonal10 (0%)0 (0%)0 (0%)1 (100%)N/AParental preference for a safe neighbourhood for kids to play (higher)Interpersonal11 (100%)0 (0%)0 (0%)0 (0%)N/AParental monitoring, not specific to OP (higher)Interpersonal21 (50%)0 (0%)0 (0%)1 (50%)N/AParental monitoring of PA (higher)Interpersonal10 (0%)0 (0%)0 (0%)1 (100%)N/AParent-perceived responsibility for child PA (higher)Interpersonal10 (0%)0 (0%)0 (0%)1 (100%)N/AParental pressure for child to be active (higher)Interpersonal10 (0%)0 (0%)0 (0%)1 (100%)N/ANumber of myopic parents (1 or 2 vs. 0)Interpersonal10 (0%)1 (100%)0 (0%)0 (0%)N/AFarmworker parent (yes vs. no)Interpersonal11 (100%)0 (0%)0 (0%)0 (0%)N/APregnancy depression scoreInterpersonal10 (0%)0 (0%)0 (0%)1 (100%)N/APregnancy screen time (higher)Interpersonal10 (0%)0 (0%)0 (0%)1 (100%)N/ARecent parental weight loss surgery (yes vs. no)Interpersonal10 (0%)1 (100%)0 (0%)0 (0%)N/ASessions with parents with additional resources (e.g., specific advice and community guide) promoting OP (yes vs. no)Interpersonal33 (100%)0 (0%)0 (0%)0 (0%)+Newsletter or booklet intervention on healthy eating and PA (yes vs. no)Interpersonal11 (100%)0 (0%)0 (0%)0 (0%)N/ACounselling intervention to reduce TV viewing (yes vs. no)Interpersonal10 (0%)0 (0%)0 (0%)1 (100%)N/ALifestyle Triple P intervention (focused on nutrition, PA, and positive parenting strategies; yes vs. no) ^a^Interpersonal11 (100%)0 (0%)0 (0%)0 (0%)N/AIntervention—structured weekend outdoor activities and incentives for children to increase steps counts via pedometers (yes vs. no)Interpersonal10 (0%)0 (0%)1 (100%)0 (0%)N/AIntervention—parent-based prevention program after parental bariatric surgery (yes vs. no)Interpersonal10 (0%)0 (0%)0 (0%)1 (100%)N/AIntervention—pediatrician outdoor PA prescriptionInterpersonal10 (0%)0 (0%)0 (0%)1 (100%)N/AParent-perceived interactions among neighbours (higher)Community10 (0%)0 (0%)0 (0%)1 (100%)N/AParent-perceived change in interactions among neighbours (increase)Community11 (100%)0 (0%)0 (0%)0 (0%)N/AParent-perceived people out and about in the neighbourhood or changes in the number of people out and about (more)Community10 (0%)0 (0%)0 (0%)1 (100%)N/AParent-perceived low crime rate in neighbourhood (increase)Community10 (0%)0 (0%)0 (0%)1 (100%)N/AParent-perceived change in crime rate in neighbourhood (lower)Community11 (100%)0 (0%)0 (0%)0 (0%)N/ASchool type (private vs. public)Community10 (0%)0 (0%)0 (0%)1 (100%)N/ASchool shift (afternoon vs. morning)Community10 (0%)0 (0%)0 (0%)1 (100%)N/AParent-perceived social capital (higher)Community11 (100%)0 (0%)0 (0%)0 (0%)N/AYouth-perceived neighbourhood support (higher)Community10 (0%)0 (0%)0 (0%)1 (100%)N/AParent-perceived neighbourhood friendliness or attractiveness for children (higher)Community10 (0%)0 (0%)0 (0%)1 (100%)N/ASchool-based sun safety intervention discouraging outdoor activities at certain times of the day (yes vs. no)Community30 (0%)2 (66.7%)0 (0%)1 (33.3%)-Preschool interventions focused on health behaviours and stress management/self-regulation (yes vs. no)Community20 (0%)0 (0%)0 (0%)2 (100%)N/ASchool curriculum intervention to increase OP (yes vs. no)Community31 (33.3%)0 (0%)0 (0%)2 (66.7%)0Parent-perceived access to local destinations for PA (higher or increase over time)Built environment20 (0%)0 (0%)0 (0%)2 (100%)N/AParent-perceived neighbourhood upkeep (higher or increase over time)Built environment10 (0%)0 (0%)0 (0%)1 (100%)N/AParent-perceived back or front yard size (larger)Built environment20 (0%)0 (0%)0 (0%)2 (100%)N/AParent-perceived change in large back or front yard size (increase)Built environment10 (0%)0 (0%)0 (0%)1 (100%)N/ALiving in a cul-de-sac vs. on a through street (or moving to a cul-de-sac)Built environment10 (0%)0 (0%)0 (0%)1 (100%)N/AParent-perceived traffic on neighbourhood streets (low or decrease)Built environment10 (0%)0 (0%)0 (0%)1 (100%)N/AParent-perceived parks and open spaces nearby (higher or increase)Built environment10 (0%)0 (0%)0 (0%)1 (100%)N/AParent-perceived sidewalks through neighbourhood (higher or increase)Built environment10 (0%)0 (0%)0 (0%)1 (100%)N/AParent-perceived neighbourhood functionality (higher)Built environment10 (0%)0 (0%)0 (0%)1 (100%)N/AParent-perceived neighbourhood attractiveness (higher)Built environment10 (0%)0 (0%)0 (0%)1 (100%)N/AParent-perceived traffic safety (higher)Built environment10 (0%)0 (0%)0 (0%)1 (100%)N/ALiving in suburban vs. traditional neighbourhoodBuilt environment10 (0%)0 (0%)0 (0%)1 (100%)N/ALiving in an urban vs. rural areaBuilt environment10 (0%)1 (100%)0 (0%)0 (0%)N/AAmbient ozone concentration (higher)Natural environment10 (0%)0 (0%)0 (0%)1 (100%)N/ASeason (warmer) ^b^Natural environment77 (100%)0 (0%)0 (0%)0 (0%)+Weather perceived as a barrier by parent (higher)Natural environment10 (0%)0 (0%)0 (0%)1 (100%)N/ATime of the week (weekdays vs. weekend days)Chronosystem11 (100%)0 (0%)0 (0%)0 (0%)N/AStreet closure intervention (yes vs. no)Policy11 (100%)0 (0%)0 (0%)0 (0%)N/ACOVID-19 restrictions (adoption)Policy30 (0%)3 (100%)0 (0%)0 (0%)-COVID-19 restrictions (removal)Policy11 (100%)0 (0%)0 (0%)0 (0%)N/ANote: BMI: body mass index; OP: outdoor play; OT: outdoor time; PA: physical activity. ^a^ The intervention effect was significant at 12 months and close to significance at 4 months (*p* = 0.063; Cohen’s *d* = 0.56) [60], so the intervention effect was deemed positive. ^b^ Sääkslahti et al. [45] also found significant difference between spring and fall seasons, but it is not included in this table because it was not clear which season was warmer. Due to the small number of studies that were not rated as “weak” based on the Effective Public Health Practice Project tool (see Table 5), the risk of bias remains high for most potential determinants.
ijerph-20-01328-t004_Table 4Table 4Moderators and mediators examined in relation to outdoor time.Author (Year)Moderators or Mediators ExaminedResultsAvol (1998) [50]Asthma status (healthy, “wheezy”, or asthmatic) as potential moderatorAsthma status did not moderate the relationship between peak hourly ambient ozone (O_3_) concentration and OT (all *p* > 0.10).French (2013) [83]Gender and ethnicity as potential moderatorsGender and ethnicity did not moderate the relationship between age and outdoor time (all *p* ≥ 0.2).Händel (2017) [38]Sex, age, mother’s BMI, mother’s education, and mother’s PA level were examined as potential moderators of the effect of the intervention.No significant interactions were found (all *p* ≥ 0.2).Handy (2008) [61]The presence of children aged 6 to 12 was examined as a potential moderator between 12 perceived neighbourhood characteristics ((1) nearby amenities; (2) neighbourhood upkeep; (3) large back yard; (4) large front yard; (5) living in a cul-de-sac rather than on a through street; (6) low traffic on neighbourhood streets; (7) parks and open spaces nearby; (8) sidewalks through neighbourhood; (9) lots of interaction among neighbours; (10) lots of people out and about in the neighbourhood; (11) low crime rate in neighbourhood; (12) safe neighbourhood for children to play) and weekly frequency of OP.The only significant interaction in longitudinal analyses was between the change in cul-de-sac and the presence of children aged 6–12 in the household (β = 0.170; *p* = 0.014), suggesting that the presence of cul-de-sacs is supportive of OP for younger children.Li (2021) [63]Child age, sex, and household income as potential moderatorsThe effect of limiting the number of visitors on OT was significant in children < 5 years (β = −9.94, 95% CI = −17.18, −2.71, *p* = 0.01), but not in older children. The effect of practising physical distancing was significant for children for families earning ≥ CAD 80,000 (β = −5.00, 95% CI: −9.47, −0.53, *p* = 0.03), but not in those earning less. Age, sex, and household income did not moderate any other associations.Miller (2017) [86]Gender and parental monitoring as potential moderatorsThe effect of youth-perceived neighbourhood danger at Time 1 on OT at Time 2 was moderated by both gender (coefficient = −0.0718, SE = 0.1392, t = −0.5157) and parental monitoring at Time 1 (coefficient = −0.0157, SE = 0.0195, t = −0.8077). The effect of youth-perceived neighbourhood support at Time 1 on OT at Time 2 was also moderated by parental monitoring (coefficient = −0.1153, SE = 0.1130, t = −1.0200) and gender (coefficient = −0.9064, SE = 0.8646, t = −1.0484). The author reported that “youth with lower levels of parental monitoring perceived their neighbourhood to be more supportive and spent more time outside. In contrast, youth with high parental monitoring, who perceived their neighbourhood to be more dangerous, spent less time outside”. The author did not clearly discuss the moderating effect of gender. Remmers (2014a) [70]Perceived responsibility regarding child PA as potential moderatorPerceived responsibility moderated the effect of perceived functionality on OP (Table 3). When stratified, functionality was related to more OP in children of parents with high perceived responsibility (β = 0.04; 95% CI = −0.07, 0.15), while among parents with low responsibility, functionality was related to less OP (β = −0.03; 95% C = −0.09, 0.04). Traffic safety interacted with perceived responsibility, but this effect was only significant after adjustment for main effects (Model 3) and became non-significant after adjustment for the other interaction (Model 4). Stratification showed that the association between traffic safety and OP was marginally stronger when parents perceived high (β = 0.10; 95% CI = −0.03, 0.23) vs. low responsibility (β = 0.06; 95% CI = −0.003, 0.12).Schneor (2021) [74]Full vs. partial COVID-19 lockdown as potential moderatorA larger reduction in OT was observed with a full lockdown vs. a partial lockdown (−93 vs. −30 min/day; *p* = 0.008).Shah (2017) [46]Age as potential moderatorAge moderated the association between future risk of myopia and OT (*p* = 0.002), such that, before the age of 4, there was little difference in OT between children who later became myopic and those who remained non-myopic. Between the ages of 4 and 8.5 years, children who later became myopic spent progressively less time outdoors than their peers who remained non-myopic (0.1 SD unit per day difference in OT by age 8.5 years).van Grieken (2014) [77]Sex as potential moderatorThe interaction between sex and group was significant (*p* = 0.019). Males in the intervention group had a change in OP of −13.99 min/day (95% CI: −46.11, 18.13) vs. 31.65 min/day for females (95% CI: 4.32, 58.98).Van Stralen (2012) [78]This study investigated several hypothesized personal and environmental mediators, which included perceived pros and cons, social pressure, social support, social modelling, self-efficacy, planning skills, perceived barriers, enjoyment, and habit strength related to OP.No statistically significant intervention effects on potential mediators were seen at Time 2; thus, criteria for mediation were not satisfied. Watowicz (2012) [87]Length of time since parental weight loss surgery as potential moderatorThere were no differences in reported lifestyle behaviours (including OP) in an analysis of the subset of subjects (*n* = 33) for whom length of time since surgery was available (by Watowicz et al.).Xu (2016) [49]Time as a potential moderator of sleep patterns (bedtime, sleep duration, sleep latency, sleep > 10 h/day, and waking at night)Interactions were excluded from the final model, as they were not statistically significant (*p* > 0.05).Note: OP: outdoor play; OT: outdoor time; PA: physical activity.
ijerph-20-01328-t005_Table 5Table 5Summary of the quality assessment using the Effective Public Health Practice Project Tool [25].Lead Author (Year)Selection BiasStudy DesignConfoundersBlindingData Collection ToolsWithdrawalsSummary RatingArcury (2017) [34]ModerateModerateModerateN/AWeakStrongModerateAvol (1998) [50]WeakModerateModerateN/AWeakStrongWeakBacha (2010) [51]ModerateModerateStrongN/AWeakModerateModerateBagordo (2017) [52]WeakModerateModerateN/AWeakModerateWeakBuller (2020) [53]WeakStrongWeakWeakWeakStrongWeakCameron (2019) [35]WeakModerateModerateN/AWeakModerateWeakChristiana (2017) [54]ModerateModerateWeakWeakWeakStrongWeakCleland (2008) [55]WeakModerateStrongN/AStrongWeakWeakCleland (2010) [56]WeakModerateStrongN/AStrongWeakWeakCortinez-O’Ryan (2017) [57]ModerateStrongWeakWeakStrongStrongWeakDavison (2011) [36]WeakModerateModerateWeakModerateModerateWeakDunton (2007) [81]WeakModerateWeakN/AModerateWeakWeakEssery (2008) [37]WeakStrongStrongWeakWeakStrongWeakEvenson (2018) [82]WeakModerateWeakN/AStrongStrongWeakFlynn (2017) [58]WeakModerateWeakWeakWeakWeakWeakFord (2002) [59]WeakStrongStrongWeakWeakStrongWeakFrench (2013) [83]ModerateModerateModerateN/AWeakWeakWeakGerards (2015) [60]ModerateStrongStrongWeakWeakModerateWeakGopinath (2013) [84]ModerateModerateStrongWeakWeakModerateWeakHändel (2017) [38]WeakStrongStrongWeakWeakWeakWeakHandy (2008) [61]WeakModerateModerateN/AWeakModerateWeakHe (2015) [12]StrongStrongModerateWeakWeakStrongWeakHesketh (2015) [39]StrongStrongWeakWeakWeakStrongWeakHnatiuk (2013) [40]ModerateModerateWeakN/AWeakModerateWeakHonda-Barros (2019) [41]StrongModerateStrongN/AWeakModerateModerateHuang (2021) [42]ModerateModerateStrongN/AWeakModerateModerateKemp (2022) [62]ModerateModerateWeakN/AWeakStrongWeakLi (2021) [63]WeakModerateStrongN/AWeakWeakWeakLi (2022) [43]ModerateModerateStrongN/AWeakWeakWeakLin (2017) [85]WeakModerateModerateN/AWeakModerateWeakLumeng (2017) [44]ModerateStrongStrongWeakModerateStrongModerateMiller (2017) [86]WeakModerateWeakN/AWeakStrongWeakMilne (2000) [64]StrongStrongModerateWeakWeakStrongWeakMilne (2007) [65]ModerateStrongModerateWeakWeakModerateWeakNigg (2021) [66]ModerateModerateWeakN/AWeakWeakWeakNgo (2009) [67]WeakStrongStrongWeakModerateStrongWeakNordvall-Lassen (2018) [68]ModerateModerateStrongN/AWeakModerateModerateOstrin (2018) [69]WeakModerateWeakN/AStrongStrongWeakRemmers (2014a) [70]ModerateModerateStrongN/AWeakStrongModerateRemmers (2014b) [71]WeakModerateStrongN/AWeakWeakWeakSääkslahti (2004) [45]ModerateStrongWeakWeakWeakModerateWeakSadeh-Sharvit (2020) [72]WeakModerateWeakN/AModerateModerateWeakSanchez-Tocino (2019) [73]WeakModerateModerateN/AWeakModerateWeakSchneor (2021) [74]WeakModerateWeakN/AStrongModerateWeakShah (2017) [46]ModerateModerateModerateN/AWeakWeakWeakShepherd-Baniga (2014) [75]WeakModerateWeakN/AWeakStrongWeakSum (2022) [76]WeakModerateWeakN/AWeakWeakWeakTandon (2019) [47]WeakStrongModerateWeakWeakStrongWeakThiering (2016) [48]ModerateModerateModerateN/AWeakWeakWeakVan Griecken (2014) [77]WeakStrongModerateStrongWeakWeakWeakVan Stralen (2012) [78]ModerateStrongStrongStrongWeakStrongModerateWalker (2021) [79]WeakModerateWeakN/AWeakStrongWeakWatowicz (2012) [87]ModerateModerateModerateN/AWeakModerateModerateWolters (2022) [80]WeakModerateWeakN/AWeakStrongWeakXu (2016) [49]WeakModerateModerateN/AModerateWeakWeakNote: Blinding was rated as non-applicable (N/A) for observational studies given the absence of intervention.


## 4. Discussion

Our systematic review summarized previous longitudinal and intervention studies examining the determinants of OT in children and youth. Overall, 119 determinants spanning the social-ecological model were examined across the 55 included studies (including 35 observational and 20 intervention studies). Illustrating the rapid growth in this field of research, the largest and most recent previous review on this topic included only 12 longitudinal studies [21]. Although we found that few potential determinants were included in a sufficient number of studies to draw conclusions about the consistency of associations, we identified some consistent positive or negative determinants of OT at multiple levels of influence.

### 4.1. Individual Level

About half of the included studies found that boys accumulated more OT than girls, with other studies reporting no gender differences. When examining findings by age group, all three cohorts that began in adolescence found that boys spent more time outdoors than girls. In the broader literature on correlates of PA, boys are usually more active than girls [8,22], with longitudinal studies suggesting that the age-related decline in PA tends to begin earlier in girls [89]. Hence, additional efforts may be needed to promote OT among girls. In our review, five studies examined sex or gender as potential moderators, and two found significant interactions. Notably, the intervention by van Grieken et al. [77] achieved a substantial increase in females’ (but not in males’) OT. Our findings suggest that the determinants of OT may vary according to sex and gender, and a better understanding of such differences could help guide future interventions.

Like the previous review by Lee et al. [21], we initially found mixed associations between age and OT. However, when considering the direction of age-related *changes* in OT by age group, we noted that increases were only reported in studies beginning in early childhood (≤5 years old), whereas decreases were common among older cohorts. These observations suggest that the relationship between age and OT may be curvilinear. If confirmed by future research, this would suggest a need for interventions to promote sustained engagement in outdoor activities in an effort to minimize the well-known age-related decline in PA [8,22,89] and increase in myopia prevalence [12]. In addition to promoting OT, interventions with youth may need to address important barriers to outdoor activities. For example, a large Canadian mixed-methods study indicated that the addictive nature of electronic screen devices and the belief that being indoors is safer and more comfortable were key factors associated with reduced connection to nature [90]. Echoing these findings, a quantitative study with Ecuadorian children found that connection to nature was negatively associated with screen time [91]. Conversely, connection to nature and OT are both associated with PA [5,6,7,8,91], suggesting that increasing outdoor activities could improve individual health [1,2] while helping to address the United Nations’ Sustainable Development Goals [17,92].

All three studies that examined the association between OT at baseline and subsequent OT found positive associations. These findings are consistent with previous research in the fields of PA epidemiology and health psychology, showing that past behaviour is an important determinant of current behaviour [93,94,95,96]. In this regard, our findings suggest that promoting OT may yield lasting benefits, though intervention studies are needed to test this hypothesis.

### 4.2. Interpersonal Level

The important role of parents in supporting their child’s PA is well established [97,98]. In our review, different aspects of parental influence (e.g., encouragement, co-participation, attitude, and intention) were examined as potential determinants of OT in no more than two studies, precluding strong conclusions. Yet, we found preliminary evidence that parent participation in PA/OP [41,69] and parental habit strength to increase OP [70] were associated with higher OT among primary school children. Ostrin et al. [69] notably reported a strong correlation within child–parent dyads in device-measured outdoor light exposure (i.e., lux) (*r* = 0.76). These findings are consistent with previous research showing consistent positive correlations in PA within parent–child dyads—especially in studies using device-based measures of PA [98].

In general, we found insufficient or inconclusive evidence regarding the associations between parental socio-demographic variables (e.g., education, income, occupation, and ethnicity) and OT (Table 3). Mostly relying on cross-sectional findings, previous reviews concluded that children from ethnic minorities and more educated parents played outside less [19,21]. It is also important to bear in mind that, in line with social-ecological theory [22], parental socio-demographic variables may interact with other levels of influence. Such interactions were not investigated in previous reviews, and we found only one study that examined income as a potential moderator [63]. Hence, more research is needed to unpack the potential role(s) of socio-demographic variables in influencing OT, as this may help in tailoring interventions to different groups.

Extending previous reviews, we were able to include 20 intervention studies. We found that all three interventions that included sessions with parents augmented by additional resources to promote outdoor activities (e.g., specific advice and community guide) were associated with higher OT [36,45,58]. While promising, two of these studies did not include a control group [36,58], and thus, further research is needed to confirm these findings.

Within the realm of interpersonal influences, nature/outdoor PA prescriptions from pediatricians and other health professionals, such as the US National ParkRx Initiative (https://www.parkrx.org/, accessed on 7 December 2022) and the Canadian PaRx (https://www.parkprescriptions.ca/, accessed on 7 December 2022), are gaining popularity to address physical inactivity and excessive screen time [88]. One outdoor PA prescription intervention met our inclusion criteria, but it was limited by a small sample size, seasonal changes, and a crude outcome measure [54]. Conversely, pilot studies from Kondo and colleagues’ [88] narrative review of nature prescription programs found positive intervention effects on the park visit frequency [99,100] but did not measure OT per se. Notwithstanding the potential benefits of park prescriptions, the currently insufficient evidence of effectiveness represents a barrier for healthcare providers to dedicate their limited time [88]. This underscores a need for larger outdoor activity prescription interventions using stronger study designs and measures.

### 4.3. Community Level

In contrast, school-based sun safety interventions that discouraged OT during midday were generally associated with lower OT [53,65]. Well-intended public health campaigns focusing on different issues have collectively recommended keeping children indoors for most of the day to avoid (1) sun exposure between 10:00 am and 4:00 pm to prevent skin melanoma; (2) ozone exposure, which tends to be highest in commuting periods immediately before and after school; and (3) exposure to insect-borne diseases from dusk to dawn [7]. Notwithstanding these issues, researchers have called for a better balance between risks and benefits and identified often overlooked risks associated with being indoors, including physical inactivity, excess screen time, an increased risk of myopia, exposure to an obesogenic food environment, indoor air pollution, allergens, and cyberpredators [7,14]. Parents, educators, pediatricians, and other health professionals could play an important role in reframing perceptions of risks associated with OT [14,101].

### 4.4. Natural Environment Level

We found that children consistently spent more time outside during the warmer months. These findings are consistent with previous reviews on PA and OT [21,102]. While seasonal variations cannot be modified in the short term, interventions could be developed to promote outdoor activities in colder seasons with appropriate clothing. Such interventions may be particularly useful for new or recent immigrants who may be at higher risk of physical inactivity [103].

### 4.5. Policy Level

Only four studies examined policy-level determinants, but all of them found significant associations with OT. Cortinez-O’Ryan et al. [57] found that an evening street closure intervention occurring twice a week for 12 weeks was associated with significant increases in measures of frequency and duration of OP. In contrast, all three studies that examined the effect of COVID-19 restrictions found a significant decrease in OT. Furthermore, in a small sample of Israeli boys, Schneor et al. [74] reported that the effect of a full lockdown was greater than that of a partial lockdown and that device-measured OT returned to pre-pandemic levels after restrictions were removed. Our findings are consistent with reviews that reported large decreases in PA and increases in screen time during the pandemic [9,104]. Given the potential of PA to reduce the risk and severity of COVID-19 infections [105] and the lower risk of transmission in outdoor environments, more efforts should be invested in promoting OT and PA during future pandemics.

### 4.6. Limitations of Included Studies

Based on the EPHPP tool, quality was rated as “weak” for most included studies. It is worth noting that some components of the EPHPP are rated quite severely [27,30]. For example, the withdrawals component is rated as “weak” whenever attrition exceeds 40%, regardless of the follow-up duration. Similarly, a participation rate below 60% is considered “weak” for the selection bias component, regardless of the complexity and/or length of the study. The blinding of participants was impossible or impractical in most intervention studies. Hence, the summary quality ratings should be interpreted with caution. That said, given the large number of “weak” ratings for data collection tools, future studies should use measures of OT that have been shown to be reliable and valid. In our review, most studies used child or parent reports and did not provide information about the psychometric properties of their measures. Some studies also appeared underpowered to examine determinants of OT. Finally, most of the studies were conducted in high-income countries, so the transferability of the findings to lower- and middle-income countries is unclear.

### 4.7. Strengths and Limitations of the Review

The strengths of the review include the focus on longitudinal studies that can establish a temporal sequence, the consideration of the entire pediatric population, and the robust screening and quality assessment process. Furthermore, we were able to consider articles published in English, French, Japanese, and Spanish, although we may still have missed articles published in other languages. Given that most potential determinants have been examined in only one or two studies, we could not conduct the planned analysis of determinants stratified by gender and age group. Similarly, the heterogeneity of the exposure and outcome measures among the included studies precluded meta-analyses. Future studies examining the same potential determinants are needed to draw stronger conclusions. We also noted that OT was measured in a variety of ways, with some studies focusing only on OP, others on outdoor PA, and others on all activities that took place outdoors (Table 1 and Table 2). Standard definitions of terms related to OT and play have recently been proposed [18], and their uptake in future studies could help minimize methodological heterogeneity.

## 5. Conclusions

Previous studies have examined 119 potential determinants of OT representing multiple levels of influence of the social-ecological model, but few determinants have been examined frequently enough to draw strong conclusions. Nevertheless, we found consistent evidence that children spend more time outdoors in warmer seasons, that OT at baseline predicted subsequent OT, and that COVID-19 restrictions and sun safety interventions discouraging midday outdoor activities were associated with less OT. The association between age and OT appears to be curvilinear, with most increases reported in early childhood, followed by decreases in late childhood and adolescence. About half of studies examining gender differences reported that boys spent more time outdoors (especially in adolescence), and the remainder found no differences. Interventions that include both parent sessions and additional resources to promote OT seem promising, but more robust research is needed to confirm their effectiveness. Given that most included studies were rated as “weak”, there remains a need for stronger studies to improve the quality of evidence. Longitudinal studies investigating the determinants of OT among underrepresented populations, notably adolescents and children and youth living in low- and middle-income countries, are also needed. Finally, future studies should consider using both subjective and device-based measures of OT, such as accelerometers and other devices equipped with light sensors that can reduce social desirability and recall biases associated with subjective measures [69,74,106].

## Figures and Tables

**Figure 1 ijerph-20-01328-f001:**
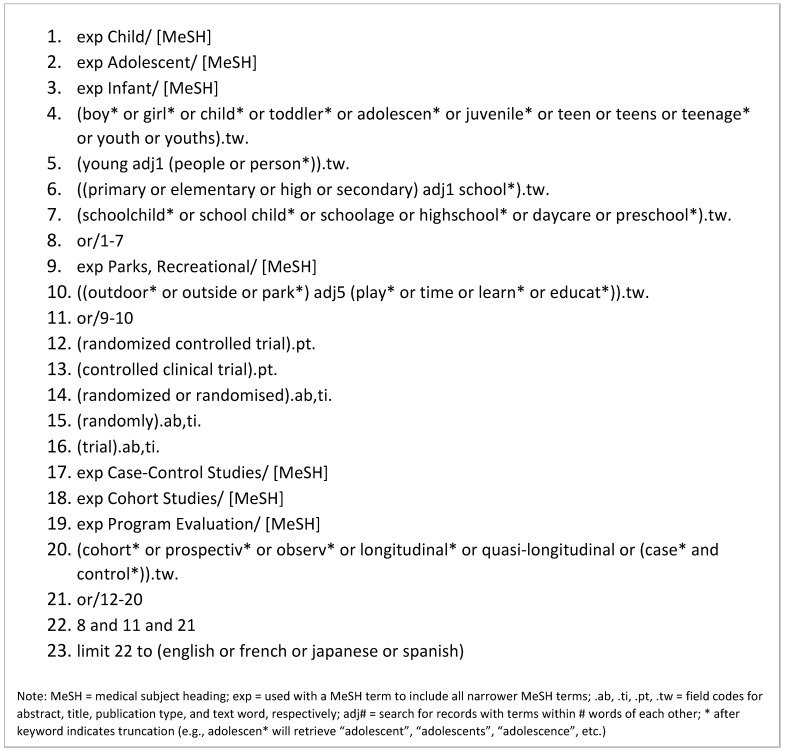
MEDLINE search strategy.

**Figure 2 ijerph-20-01328-f002:**
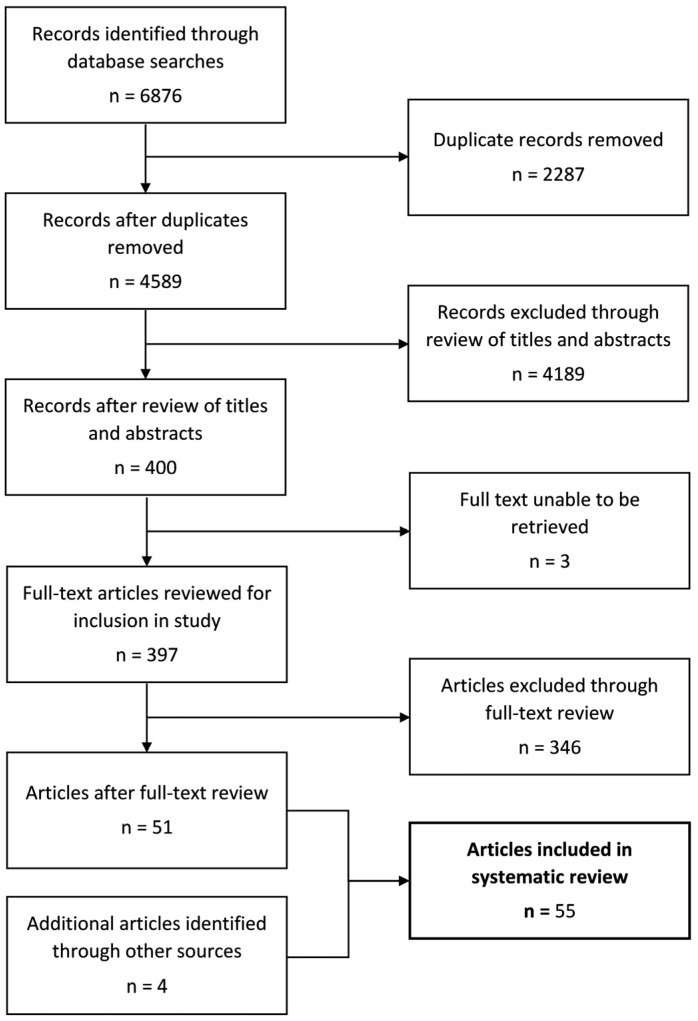
Flow diagram for the review process.

**Table 1 ijerph-20-01328-t001:** Descriptive characteristics of included studies.

Lead Author (Year)	Location	N	Age/Grade at Baseline (Mean ± SD or Range)	Gender	Setting	Follow-Up Length	Measure of Outdoor Time
**Studies Beginning in Early Childhood (<5 Years)**							
Arcury (2017) [34]	United States (North Carolina)	221	2–3 years	52% girls	Community	2 years	Parent-reported. 24 h recall: number of minutes child spent (1) at playground and (2) in the yard
Cameron (2019) [35]	Australia (Melbourne)	307	3.61 years (range: 3.2–4.5)	Not stated	Cohort	2 years	Parent-reported OT on an average day in last week
Davison (2011) [36]	United States (New York State)	1322	2–5 years	51.0% girls	Women and children clinics	1 year	Parent-reported OT (dichotomized as ≥60 min/day vs. less)
Essery (2008) [37]	United States (Texas)	90	3.1 ± 1.1 years	53% girls	Home	3 months	Parent-reported. Time spent in OP per day
Händel (2017) [38]	Denmark (Copenhagen)	307	4 years (median)	42.2% girls	Municipality (birth registry)	15 months	Parent-reported OP in min/week
Hesketh (2015) [39]	Australia (Melbourne)	542	3.9 ± 1.5 months	47.4% girls	Maternal and Child Health service	16 months	Parent-reported OT per day
Hnatiuk (2013) [40]	Australia (Melbourne)	206	3.5 ± 1 months	46.6% girls	First-time parent groups	5 months	Parent-reported OT in min/week
Honda-Barros (2019) [41]	Brazil (Recife, Pernambuco)	700	3–5 years	47.9% girls	Schools	2 years	Standardized interview (parent-reported OP in min/day)
Huang (2021) [42]	China (Longhua)	26,611	1 year	45.7% girls	Schools	2 years	Parent-reported frequency and duration of OT
Li (2022) [43]	China (Changsha)	953	12 months	48.4% girls	Clinics/hospitals	4 years	Parent-reported OT (hours/day)
Lumeng (2017) [44]	United States (Michigan)	697	4.1 ± 0.5 years	51% girls	Head Start programs (preschool)	7 months	Parent-reported OT on weekdays and weekend days
Sääkslahti (2004) [45]	Finland (Turku)	228	Intervention: 4.6 ± 0.5 years; control: 4.4 ± 0.4	48.2% girls	Clinic	3.5 years	Parent-reported. Diary reporting time spent in OP (hours/weekend)
Shah (2017) [46]	United Kingdom (Avon)	2833	2 years	50.2% girls	Cohort	13 years (6.5 years for OT)	Parent-reported OT per day
Tandon (2019) [47]	United States (Seattle, Washington)	82	Active play group: 4.5 ± 0.6 years; outdoor play group: 4.6 ± 0.4	56.1% girls	Preschools	12 weeks	Direct observation of outdoor activities (child-initiated and teacher-initiated)
Thiering (2016) [48]	Germany	837	Birth	49% girls	Birth cohorts	15 years	Child-reported OT: h/day
Xu (2016) [49]	Australia (Sydney)	369	2 years	50% girls	Birth cohort	3 years	Mother-reported. Hours of OP per weekday and weekend day
**Studies beginning in childhood (5–11 years)**							
Avol (1998) [50]	United States (Southern California)	195	10–12 years	48.7% girls	Cohort (hospital)	Mid-spring–late summer (~4–5 months)	Child-reported (diary). Location recorded hourly for four days
Bacha (2010) [51]	United States (10 sites)	868	Grade 3	50.8% girls	Birth cohort	2 years (for main exposure)	Child-reported OT in the neighbourhood on weekdays between school dismissal and 6 pm (dichotomized as any vs. none)
Bagordo (2017) [52]	Italy (5 towns)	1164	6–8 years	49.1% girls	Schools	~5 months (winter 2014–2015 to late spring 2015)	Parent-reported OP for >1 h per day (yes/no)
Buller (2020) [53]	United States (California)	1758	8.16 ± 2.04 years	49.1% girls	Schools	20 months ^a^	Parent-reported. OT between 10 am and 4 pm during the week (none, 30 min or less, or 31 min or more)
Christiana (2017) [54]	United States (Western North Carolina)	70	5–13 years	44.3% girls	Pediatric clinics	3 months	Parent-reported frequency of OT
Cleland (2008) [55]	Australia (Melbourne)	548	Two cohorts: 5–6 years and 10–12 years	53% girls	Schools	3 years	Parent-reported OT on weekdays and weekend days for warmer and cooler months
Cleland (2010) [56]	Australia (Melbourne)	421	Two cohorts: 5–6 years and 10–12 years	51.5% girls	Schools	5 years	Parent-reported OT on weekdays and weekend days for warmer and cooler months
Cortinez-O’Ryan (2017) [57]	Chile (Santiago)	100	4–12 years	51% girls	Neighbourhood	12 weeks	Parent-reported frequency and duration of OP
Flynn (2017) [58]	United States (Southeast region)	27	10.7 ± 3.3 years	51.9% girls	Neighbourhoods	4 weeks	Parent-reported. Total minutes of outdoor PA/week
Ford (2002) [59]	United States (Atlanta, Georgia)	28	7–12 years	53.4% girls	Community clinic	4 weeks	Parents and children reported together: typical amount of OT per day
Gerards (2015) [60]	Netherlands (Limburg)	56	7.2 ± 1.4 years	55.8% girls	Public health services	12 months	Parent-reported. Days per week and number of hours playing outside
Handy (2008) [61]	United States (Northern California)	272	<16 years	Not stated	Neighbourhoods	1 year (retrospective)	Parent-reported. Frequency of OP in previous week
He (2015) [12]	China (Guangzhou)	1848	6–7 years	46% girls	Schools	3 years	Parent-reported OT in min/day
Kemp (2022) [62]	Australia (national)	2971	10.4 ± 0.5 years	49.2% girls	Home	2 years	Child-reported time-use diaries with a category for non-organized outdoor/nature PA in min/day
Li (2021) [63]	Canada (Toronto)	265	5.5 ± 2.5 years	47.5% girls	Clinics/hospitals	3 months	Parent-reported OT (hours and min/day)
Milne (2000) [64]	Australia (Perth)	1386	5–6 years	48% girls	Schools	17 months	Parent-reported. Average time each day that the children were outdoors between 8 am and 4 pm and between 11 am and 2 pm
Milne (2007) [65]	Australia (Perth)	1116	5–6 years	49.8% girls	Schools	4 years	Parent-reported. Average time each day that the children were outdoors between 8 and 11 AM, between 11 AM and 2 PM, and between 2 and 5 PM
Nigg (2021) [66]	Germany	570	5.3 ± 0.8 years	54.7% girls	Community	11 years	Participant-reported OP frequency in a typical week (from 0 to 7 days)
Ngo (2009) [67]	Singapore	285	6–12 years	45.9% girls	Program/community	9 months	Parent-reported OT on weekdays and weekend days (questionnaire and diary)
Nordvall-Lassen (2018) [68]	Denmark (Aarhus)	4941	9–11 years	49.6% girls	Birth cohort	9–11 years	Parent-reported OT (h/week)
Ostrin (2018) [69]	United States (Houston, Texas)	60	7.6 ± 1.8 years	40% girls	Cohort	1 year	Parent-reported OT and Actiwatch-measured ambient light exposure
Remmers (2014a) [70]	Southern Netherlands	1317	5.0 ± 0.5 years	49% girls	Cohort	2 years	Parent-reported OP. Frequency and duration in an average week for the last 4 weeks
Remmers (2014b) [71]	Netherlands	2007	5.75 ± 0.42 years	49.5% girls	Healthcare cohort	2 years	Parent-reported. Total duration of unstructured OP in an average week
Sadeh-Sharvit (2020) [72]	United States	7	5.95 ± 3.57 years	Not available	Hospital	8 weeks	Parent-reported. OP checklist
Sanchez-Tocino (2019) [73]	Spain (Valladolid and Burgos, Castilla y León)	82	10 ± 3 years	52.4% girls	Hospitals	1.5 years	Parent-reported OT. Hours/week
Schneor (2021) [74]	Israel (central)	19	10.2 ± 0.9 years	0% girls	Clinics	21 months	Actiwatch-measured ambient light exposure
Shepherd-Banigan (2014) [75]	United States (Eastern Washington State)	99	6–12 years (median: 9.5)	48% girls	Farming setting	9 months	Parent-reported. Daily diary reporting children’s OT
Sum (2022) [76]	Singapore	604	7.1 ± 3.6 years	50.8% girls	Clinics	3–5 months (retrospective)	Parent-reported frequency of OP or exercise
Van Griecken (2014) [77]	Netherlands	293	5.8 ± 0.4 years	61.9% girls	Healthcare centres	2 years	Parent-reported (dichotomized as playing outside <1 h vs. ≥1 h per day)
Van Stralen (2012) [78]	Netherlands (Amsterdam)	600	9.8 ± 0.7 years	51% girls	Schools	20 months	Child-reported frequency of OP
Walker (2021) [79]	United States (Texas)	13	5–10 years	46% girls	University (OP room)	8 weeks	Parent-reported OT (hours/day) on weekdays and weekend days
Wolters (2022) [80]	Belgium, Cyprus, Estonia, Germany, Hungary, Italy, Spain, and Sweden	2094	6.2 ± 1.8 years	49% girls	School	6 years	Child- or parent-reported OT (hours/day) for weekdays and weekend days
**Studies beginning in adolescence (12–17 years)**							
Dunton (2007) [81]	United States (Southern California)	524	14.5 ± 0.5 years	49% girls	Schools	4 years	Electronic ecological momentary assessment: proportion of entries in outdoor context
Evenson (2018) [82]	United States (California and Minnesota)	265	Grades 10–11	100% girls	Parks	1 year	GPS-measured. Minimum park visit duration of 3 min to count as time spent in parks
French (2013) [83]	Australia (Sydney)	1739	6.7 (younger cohort) and 12.7 (older cohort)	47.3% girls	Cohort	5–6 years	Parent-reported (and child-reported if 12+ years old). Sum of weekly time spent in outdoor leisure and sport
Gopinath (2013) [84]	Australia (Sydney)	752	12.7 years at baseline	53.3% girls	Cohort	5 years	Sum of youth-reported time spent in different outdoor sporting activities in an average week
Lin (2017) [85]	China (Beijing)	217	8.4 ± 1.1 and 14.2 ± 1.7 years for the primary and secondary students	51.2% girls	Clinics	3 years	Child-reported (parental help if needed). Sum of hours spent in outdoor sports and leisure after school
Miller (2017) [86]	United States (Chicago, Illinois)	250	Mean age of 12 years	59% girls	Neighbourhoods	1 year	Child-reported (ecological momentary assessment). Location reported ~7 times throughout the day.
Watowicz (2012) [87]	United States (Midwest)	135	12.8 ± 2.8 years	60% girls	Hospital (patients of a pediatric weight management centre)	45 months (range = 8 to 86 months) [retrospective]	Children and parents completed questionnaire together (OP dichotomized as <1 vs. ≥1 h/day)

Note: N = Sample size (only participants with outdoor time data were included); OP = outdoor play; OT = outdoor time; PA = physical activity; SD = standard deviation. ^a^ For the Buller et al. [33] study, the time period between pretest and post-test was not clear, but the intervention lasted 20 months.

## Data Availability

Template data collection forms can be requested via email to R.L.

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
