# Peer review of "Determinants of Outdoor Time in Children and Youth: A Systematic Review of Longitudinal and Intervention Studies"

_ijerph, 2023, doi:10.3390/ijerph20021328_

Round 1
Reviewer 1 Report
Very interesting and high-quality review. I only suggest format improvements, not content, which are as comments in the attached file.

Reviewer 2 Report
The paper is basically well written and the study was conducted in a serious and rigorous manner. The topic being investigated is though not academically interesting. More work has to be done to justify the reason for the need to understanding about OT, as its benefits of OT are quite obvious. OT in children and youth (aged 0-17 years) sounds way too broad. I do not feel like it is a homogeneous group at all. There are also too many tables and most of them are too long therefore difficult for readers to follow. Not necessary to include the main results of each study (Table 2). It is also not necessary to list out all the determinants in each study (Table 3).
Reviewer 3 Report
see file attached
